# Robust Predictors for Seasonal Atlantic Hurricane Activity Identified with Causal Effect Networks

Peter Pfleiderer[1,2,3*], Carl-Friedrich Schleussner[1,2,3], Tobias Geiger[3,4], Marlene Kretschmer[5]

[1] Climate Analytics, Berlin, Germany
[2] IRI THESys, Humboldt Universität Berlin, Berlin, Germany
[3] Potsdam Institute for Climate Impact Research, Potsdam, Germany
[4] German Meteorological Service, Climate and Environment Consultancy, Stahnsdorf, Germany
[5] University of Reading, Department of Meteorology, Reading, UK

*Correspondence to*: Peter Pfleiderer (peter.pfleiderer@climateanalytics.org)

**Abstract.** Atlantic hurricane activity varies substantially from year to year and so do the associated damages. Longer-term forecasting of hurricane risks is a key element to reduce damages and societal vulnerabilities by enabling targeted disaster preparedness and risk reduction measures. While the immediate synoptic drivers of tropical cyclone formation and intensification are increasingly well understood, precursors of hurricane activity on longer time-horizons are still not well

established. Here we use a causal network-based algorithm to identify physically interpretable late-spring precursors of seasonal Atlantic hurricane activity. Based on these precursors we construct statistical seasonal forecast models with competitive skill compared to operational forecasts. In particular, we present a skillful prediction model to forecast July to October tropical cyclone activity at the beginning of April. Our approach highlights the potential of applying causal effect network analysis to identify sources of predictability on seasonal time-scales.

## 1 Introduction

Tropical cyclones (TCs) are among the most damaging weather events in many tropical and subtropical regions (Munich Re 2020). The compound nature of tropical cyclone hazards combining heavy winds, extreme precipitation and coastal flooding contributes to their severity (Ye and Fang 2018), directly impacting societies. Furthermore, a range of secondary impacts in the aftermaths of cyclones such as displacement, loss in livelihoods or income, and health impacts are being reported (Camargo

and Hsiang 2014). Applying risk reduction measures to the direct damages of TCs is challenging and is expected to become even more so with global warming and sea level rise (Woodruff et al. 2013). Preparedness for the secondary impacts could, however, be improved if reliable forecasts of the potential risks of the upcoming hurricane season are available (Murphy et al. 2001).

Several academic institutes provide seasonal hurricane forecasts for the Atlantic basin (Klotzbach et al. 2019). The Colorado

state university was one of the first, already issuing seasonal forecasts in 1984 (Gray 1984a, b). Since then, a variety of forecasting methods are applied, ranging from purely statistical forecasts to forecasts based on numerical global climate model

simulations and hybrid approaches (Klotzbach et al. 2017, 2019). The Barcelona Super Computing Center each year collects and publishes seasonal forecasts from universities, private entities and government agencies[1].

Dynamical forecasts are based on global circulation models that simulate the climate system including tropical cyclone occurrences (Vitart and Stockdale 2001; Vecchi et al. 2014; Manganello et al. 2017). Their skill depends on their ability to represent TC genesis and development, and their capacity to forecast the large-scale circulation over the Atlantic main development region (MDR) as well as their ability to adequately represent the interaction between the two. With increasing spatial resolution, their representation of TCs improves (Roberts et al. 2020). Their ability to predict the large-scale circulation and low frequency variability can, however, remain a limiting factor for seasonal forecasts (Manganello et al. 2017).

Statistical forecast models, in contrast, are usually based on favorable climatic conditions in the region of TC formation and established teleconnections affecting cyclone activity on the basins scale (Klotzbach et al. 2017). Besides warm sea surface temperatures (SST), both the formation and intensification of TCs critically depend on low vertical wind shear (VWS) over the tropical Atlantic (Frank and Ritchie 2001; Emanuel et al. 2004). Furthermore, dry air intrusion and anticyclonic wave breaking can hamper TC formation (Hankes and Marinaro 2016). Finally, a lack of easterly African waves can lead to lower TC activity (Dieng et al. 2017; Patricola et al. 2018).

The included predictors of a statistical forecast model are often chosen based on correlation analysis and expert judgement. One major challenge in statistical forecasting is yet to select a set of skillful predictors without running into overfitting issues, implying dropping skill when applied to independent test data (Hawkins 2004).

Recently, a novel statistical forecast approach based on causal effect networks (CEN) was proposed (Kretschmer et al. 2017). In such a network, causal links between the predictand and a set of potential predictors are identified by iteratively testing for conditionally independent relationships, thereby removing spurious correlations (Runge et al. 2019). First applications have shown that statistical forecast models based on causal precursors can result in skillful forecasts as they identify relevant predictors without overfitting (Kretschmer et al. 2017; Di Capua et al. 2019; Saggioro and Shepherd 2019; Lehmann et al. 2020).

Here we apply this approach to detect remote predictors in spring of hurricane activity in the Atlantic basin from July to October. We first demonstrate the applicability of the method by constructing a forecast for the July-October accumulated cyclone energy (ACE) based on May precursors using reanalysis data. The identified precursors are well-documented drivers of hurricane activity in the Atlantic, indicating the usefulness of our approach. To increase forecast lead time, we then apply the same method to construct a forecast based on March reanalysis and obtain competitive forecast skill based on these predictors.

---

[1] https://seasonalhurricanepredictions.bsc.es/predictions

## 2 Methods

### 2.1 Data

We use tropical cyclone locations and maximum sustained wind speeds from the official WMO agency from the IBTrACS database (Knapp et al. 2010, 2018). Our main analysis is based on the fifth generation of ECMWF atmospheric reanalyses (ERA5) (Copernicus Climate Change Service (C3S) 2017). We use the monthly reanalysis data on a regular 1-degree grid for the period 1979-2018. For sensitivity testing, we also use the Japanese 55-year reanalysis (JRA55) on monthly time-scale and provided on a regular 1.25-degree grid (The Japan Meteorological Agency (JMA) 2013). As data in the pre-satellite era are less reliable (Tennant 2004), we focus on the period from 1979-2018, but we also perform sensitivity tests using the full range of the JRA55 dataset ranging from 1958-2018.

### 2.2 Accumulated Cyclone Energy (ACE)

Following Waple et al. (2002), we calculate accumulated cyclone energy (ACE) as an indicator for seasonal tropical cyclone activity:

$$ACE = 10^{-4} \sum_{all\ days} v_{max}^2 \ ,$$

ACE is accumulated for TCs within the Atlantic basin with maximal sustained wind speeds above 34 knots over all days from July-October.

### 2.3 Causal effect networks (CEN)

Causal effect networks have been introduced to statistically analyze and visualize causal relationships between different climatic processes, referred to as "actors". Specifically, spurious correlations due to indirect links, common drivers or autocorrelation effects are identified as such and removed from the network structure (Kretschmer et al. 2016; Runge et al. 2019). The remaining links can then be interpreted in a more causal way within the set of considered variables.

Here we use a two-step approach to construct causal effect networks consisting of a condition selection algorithm (PC-algorithm) and a momentary conditional independence (MCI) test. This so called PCMCI algorithm was introduced by Runge te al. (2019) and a python implementation is openly available on github.com/jakobrunge/tigramite (Runge 2014). The properties of the PCMCI algorithm including mathematical proofs and numerical tests are documented and discussed in Runge et al. (2019).

Note that this algorithm relies on several assumptions, which in real-world scenarios are likely never fully fulfilled (Runge 2018). Specifically, it requires a comprehensive sampling of potentially relevant climate signals as well as sufficient temporal coverage to ensure full representation of multi-annual to multi-decadal modes. As we are particularly restricted by the relatively short reanalysis record, we cannot exclude potential state-dependencies, e.g. on annual time scales as well as non-stationarities (Fink et al. 2010; Caron et al. 2015). As this represents a divergence from the theoretical methodological approach of causal

precursor analysis, we will therefore refer to the results of the CEN analysis as "robust precursors" acknowledging that we cannot assure *true* causality.

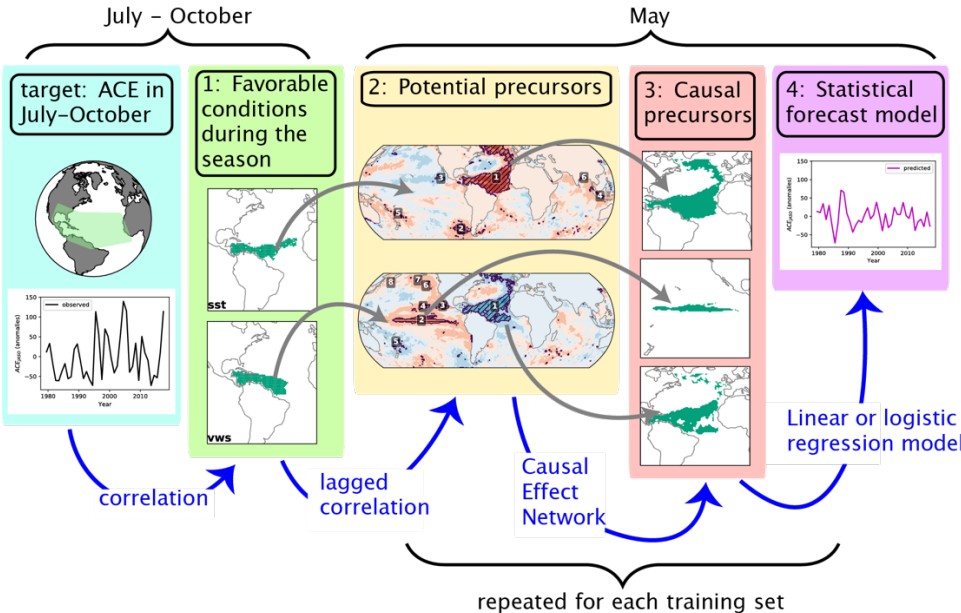

**Figure 1: Schematic overview of the four steps to build a forecast model for ACE in July-October (left): 1) The regions of interest for two favourable conditions of hurricane activity (SST and VWS) are identified. 2) For each of the favourable conditions of 1) potential precursors in May are identified by clustering the most significantly correlated grid cells within SST data. 3) Causal effect networks are used to select a sub-set of robust precursors. 4) A statistical model is built based on the identified robust precursors of 3). Steps 2)-4) are repeated for the different training sets leading to a different forecast model for each hindcasted year.**

## 2.4 Using CEN as a robust precursor selection tool to construct a statistical forecast model

We apply a CEN approach to identify robust precursors in May (and in March) of seasonal hurricane activity of the same year. Similar to Kretschmer et al. (2017), our methodology consists of 4 steps (see schematic overview in Fig. 1):

1. We first identify regions where **favorable conditions for TC** formation and intensification are most relevant. Here we use SSTs and VWS fields as established favorable conditions but without prescribing spatial patterns a priori. We then identify the regions in the tropical Atlantic that are correlated with ACE in our target region during the hurricane season (July to October).
2. We search for *potential* **precursors of the favorable conditions** identified in step 1. To do this, we calculate lagged point correlation maps of gridded SST, and mean sea level pressure (MSLP) data and cluster the most significantly correlated points into potential precursor regions. We use SSTs and MSLP as they are commonly used to describe the forcing on the atmosphere and the current location of pressure systems which in turn gives insights on the atmospheric circulation in general.
3. We identify *robust* **precursors** amongst all potential precursor regions identified in step 2 by constructing a causal effect network (CEN) using the so-called PCMCI algorithm.

4. We construct a **statistical forecast model** based on the robust precursors identified in step 3 using linear regression and logistic regression. While for the detection of potential and robust precursors (step 2 & 3) detrended anomalies of climate variables are used, the final forecast models are constructed with the raw reanalysis data.

More details including all relevant free parameters of our approach (such as significance thresholds and clustering parameters) are listed and discussed in the SI.

## 2.5 Forecast model evaluation

We evaluate the skill of our model by performing a cross-validation hindcast: For each hindcasted year, we construct a new statistical model using all years but the year we aim to hindcast as well as the two preceding years of that year. Specifically, steps 2-4 are iteratively performed for each hindcasted year (see Fig. 1). By excluding the two preceding years from the training

set, we assure that autocorrelations of up to 3 years do not leak information from the training data into the testing data. Note that despite the clear separation between training and testing data such cross-validation tests cannot guarantee reproducibility of the forecast skill in a real forecasting setting (Li et al. 2020).

## 3 Results

### 3.1 Favorable conditions for active hurricane seasons

Favorable conditions for active hurricane seasons are (among others) warm SSTs and low VWS over the western tropical North Atlantic (Fig. 2). We identify the regions where the association of SSTs and VWS on basin wide ACE is strongest by clustering most strongly correlated grid-cells (see SI for more information). These regions cover large parts of the main development region (MDR) and we call them $SST_{MDR}$ and $VWS_{MDR}$. The relationships between these variables and TC formation as well as TC intensification are well documented (Frank and Ritchie 2001).

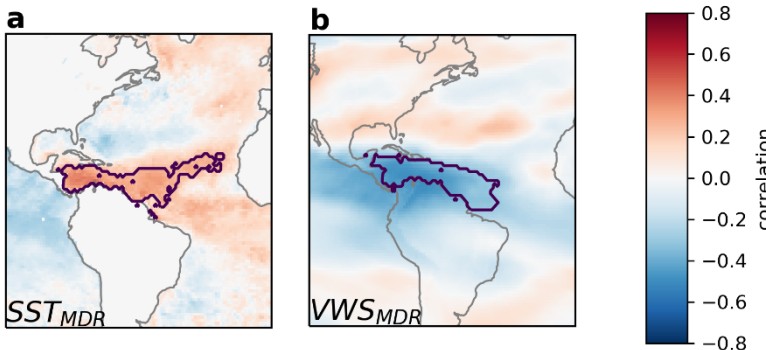

**Figure 2: Favorable conditions for high ACE in July-October. a) Point correlation between SST and basin wide mean ACE in July-October. The contour line indicates the identified region in the Atlantic basin, which consist of a cluster**

**of the 5% most significantly correlated grid-cells. Details on the definition of the region are described in the supplementary information. b) As (a) but for VWS.**

To identify **potential precursors** (step 2 in Fig. 1) of SST$_{MDR}$ and VWS$_{MDR}$ we next calculate lagged point correlation maps using the regional averages of VWS$_{MDR}$ (SST$_{MDR}$) and gridded SST data. VWS$_{MDR}$ in July-October is strongly correlated with SSTs in May in several locations. Potential precursor regions are found in the tropical Atlantic and Pacific, in the northern North Atlantic and in the northeastern Pacific (see Fig. 3a).

We then construct a **causal effect network** (step 3 in Fig. 1) with all identified potential precursors. We find that warm SSTs in the tropical Pacific and cold SSTs in the subtropical North Atlantic are *robust* precursors of strong VWS$_{MDR}$ (Fig. 3b). The signal from the Pacific resembles the Nino3.4 region and thus reflects the El Niño Southern Oscillation (ENSO) which is a well-known driver of variations in hurricane activity (Gray 1984a; Tang and Neelin 2004; Kim et al. 2009). In combination, the difference between tropical Atlantic SSTs and tropical Pacific SSTs is consistent with the hypothesis that Atlantic hurricane activity mainly depends on the temperature of Atlantic SSTs relative to the other basins (Vecchi and Soden 2007; Murakami et al. 2018).

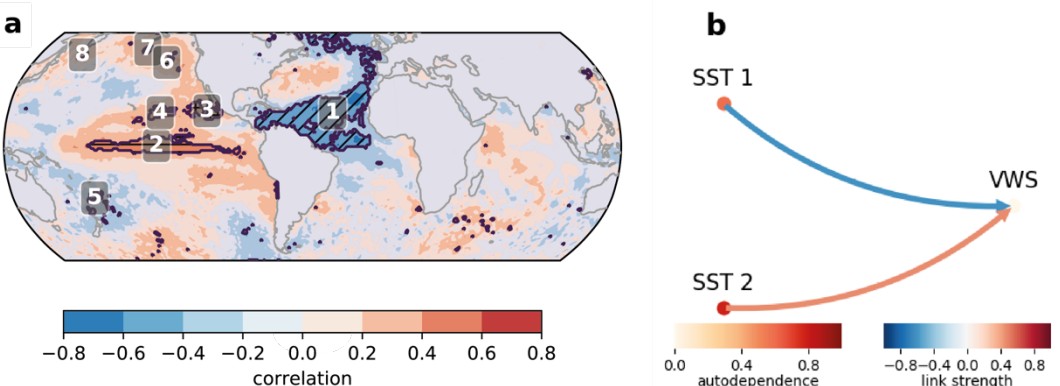

**Figure 3: Precursors of VWS$_{MDR}$ in a training set containing the years 1979-2015: a) Pointwise correlation between SSTs in May and VWS$_{MDR}$ averaged over July-October. Labels indicate clustered regions that are treated as potential precursors. b) Robust precursors of VWS$_{MDR}$ as detected by our method. The color of the arrows indicates the link strength and the color of the nodes indicates the strength of auto-dependence. The link strength (including auto-dependence of variables) is calculated following Runge et al. (2019) using partial correlations. This was shown to give a normalized measure of causal strength ranging between -1 and 1. For visualization purposes only ingoing links of VWS$_{MDR}$ are shown here, the full network is shown in Fig. S1.**

The correlation maps vary for the different training sets, partly leading to different potential precursors of $VWS_{MDR}$ (Fig. 4a).
For instance, some regions are only identified as potential precursors in some training sets (lighter shading). Nevertheless, throughout all different training sets, SSTs in the Atlantic and in the Niño3.4 region are consistently identified as robust precursors of $VWS_{MDR}$ (Fig. 4c).

A robust precursor for warm $SST_{MDR}$ in July-October is a large SST region in the North Atlantic (Fig. 4d). This region extends to the north-eastern Atlantic. The strong link of this precursor to $SST_{MDR}$ is a result of the high autocorrelation of SSTs. Furthermore it is likely, that water from north of the MDR would be advected into it during the following months (Klotzbach et al. 2019). The identified SST signals north in the subpolar Atlantic and Arctic ocean may not have a direct impact on the cyclone activity, but could also be the result of the presence of a common driver of multi-month/annual SST in the North Atlantic that cannot be resolved by our temporally limited application of the CEN method for forecasting purposes. This does not mean that the SST signal in the region does not have skill as a robust precursor, but only that no direct 'causal' pathway might be at play here.

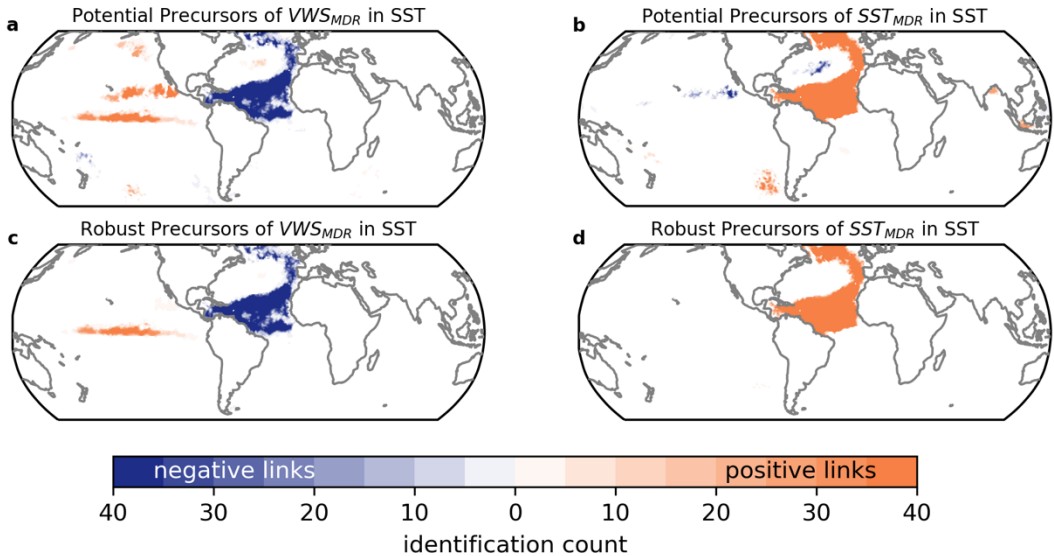

Figure 4: Potential and robust precursors of VWS_MDR (a and c) and SST_MDR (b and d) in May. Number of training sets in which a grid-cell is part of a potential (a-b) and robust (c-d) precursor region. The maximum number is 40 - the number of different training sets considered.

## 3.2 Forecast model based on May precursors

We next hindcast each years' ACE in July-October with a linear regression model (step 4 in Fig. 1) based on the absolute values of the robust precursors identified in May for the training set (containing again all years but the hindcasted year and the

two preceding years) (see Fig. 5). With a Pearson correlation coefficient of $\rho = 0.47$ (and a Spearman rank correlation coefficient of $\rho_{rank} = 0.53$), our cross-validated hindcast seems competitive with operational forecasts (from CSU, TSR and NOAA) which have $\rho < 0.4$ (see Fig. 1 in Klotzbach et al. 2019).

Our model skillfully discriminates between above and below median seasonal activity (Fig. 5b). However, the intensity of most extreme hurricane seasons is underestimated in our linear forecast model (e.g. years 1995, 2004, 2005, 2017 in Fig. 5a).

Figure 5b shows that despite this lack in sensitivity of the linear model, it can still deliver valuable information on the occurrence of above 66[th] percentile seasons.

As an addition to the linear model, we next use a logistic regression classifier to construct probabilistic forecast models. We focus on predicting the most active (above 66[th] percentile) and least active (below 33[th] percentile) seasons using the same predictors as for the linear model (Fig. 6b-c). For each year this model gives a probability of having an above 66[th] (below 33[rd])

percentile season. As it does not assume a linear relationship between predictors and predictands it might be better suited for the prediction of extreme seasons.

We evaluate the performance of the model using the Brier skill score (BSS) (Brier 1950). With a positive BSS, the result of the forecast model that gives the probability of finding an above 66[th] percentile season (Fig. 6c) is slightly superior to a climatological forecast, which would be forecasting above 66[th] percentile seasons with a probability of 33% in each year. The

reliability curve flattens out for high forecast probabilities indicating that the usefulness of this forecast is however limited. For instance, the false positive rate is 50% for seasons which are hindcasted to be an above 66[th] percentile season with a probability of 60%. Yet, seasons that are very unlikely to become particularly active are hindcasted with high confidence.

We hypothesize that the deficit to hindcast some of the most active seasons might be due to missing relevant predictors. For example, Klotzbach et al. (2018) argued that the extreme TC activity in 2017 was due to an enhanced Pacific Walker circulation

during near neutral ENSO conditions. The Pacific Walker circulation and ENSO are strongly correlated, but in 2017, forecast models using ENSO as a predictor (rather than the Walker circulation) heavily underestimated the seasonal activity (Klotzbach et al. 2018). Furthermore, it has to be noted that there is some stochastic component to TC formation which systematically limits the skill of our empirical forecast model that is based on favorable conditions for TC formation.

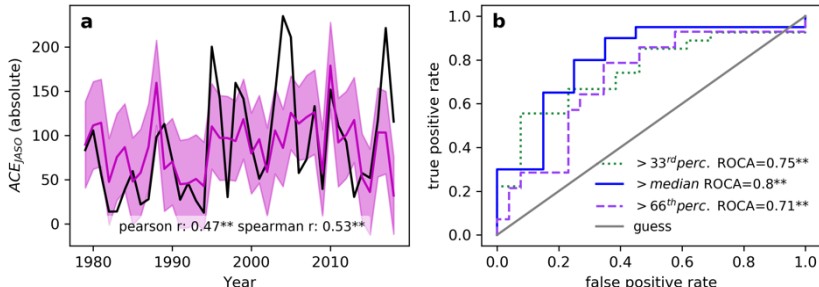


**Figure 5: Hindcast skill based on May reanalysis. a: ACE yearly aggregated over July-October (black) and based on our linear forecast model using precursors identified for the month of May (magenta). The shading corresponds to a 66% confidence interval based on the standard deviation of the model over the training periods. b: Receiver operating characteristic (ROC) curve (see SI) for different seasonal activities: above (the long-term) median seasons in blue, above**

**the 33rd percentile in green and above the 66th percentile in purple. The area under the ROC curve (ROCA) is indicated in the legend with significance levels (** - alpha=0.05, * - alpha=0.1).**

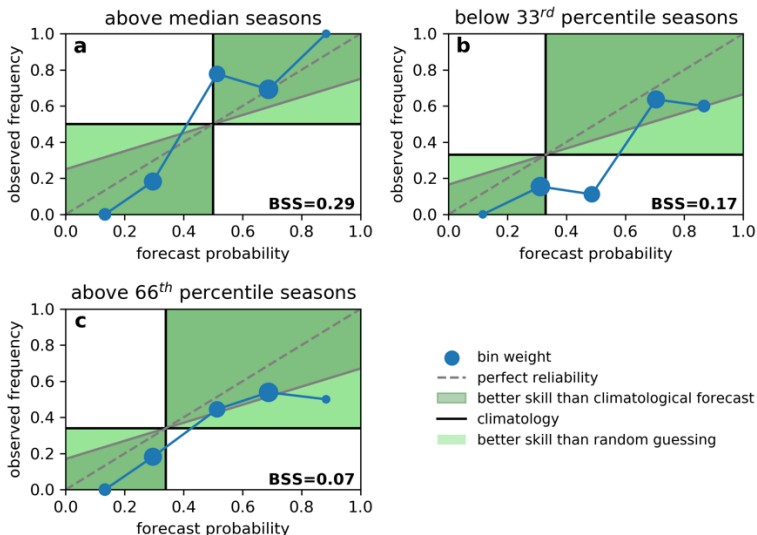

**Figure 6: Reliability diagrams for a logistic regression model on May precursors and for three types of seasonal activities: above median (a), below 33rd percentile (b), above 66th percentile (c) seasons. Dots show the mean hindcasted**

**probability versus the observed frequency of a (seasonal) event. The size of the dots indicates the relative amount of data points that contributed to a bin. A perfectly reliable forecast would lie on the diagonal (dashed gray line). Dots within the dark-green area contribute to a forecast skill improvement compared to the climatology while dots within the light-green area contribute to a forecast skill improvement compared to random guessing. The Brier Skill Score (BSS) is indicated in the lower right corner of each panel.**

### 3.3 Forecasting at longer lead times

So far, the robust precursors we detected with our data-driven approach are already well documented in the literature, providing us with confidence in the approach. We next apply the same methods to construct a forecast model based on reanalysis data in March, where existing operational forecasts show little skill (Klotzbach et al. 2017, 2019). At the end of March, it is difficult to forecast the state of ENSO for the upcoming hurricane season due to the ENSO predictability barrier (Torrence and Webster 1998; Hendon et al. 2009). Indeed, in March, the El Niño region is not identified as a robust precursor of $VWS_{MDR}$ in July-October (see Fig. S2).

We further search for robust precursors in mean sea level pressure data (MSLP). To avoid spurious effects at this long-time lag on atmospheric time scales, we adjust our criteria to yield large-scale precursors and cluster the 7.5% most significantly correlated grid cells (instead of 5% elsewhere) into large-scale precursor regions (see SI for more details).

We identify potential precursor regions in both hemispheres (Fig 7a). As robust precursors, a high-pressure system over the southern Indian Ocean and a low-pressure system eastward of New Zealand are identified in nearly all training sets (Fig 7c). For $SST_{MDR}$, autocorrelation still plays an important role and, as for the May forecast, a larger area in the North Atlantic remains a robust precursor (see Fig 7 d).

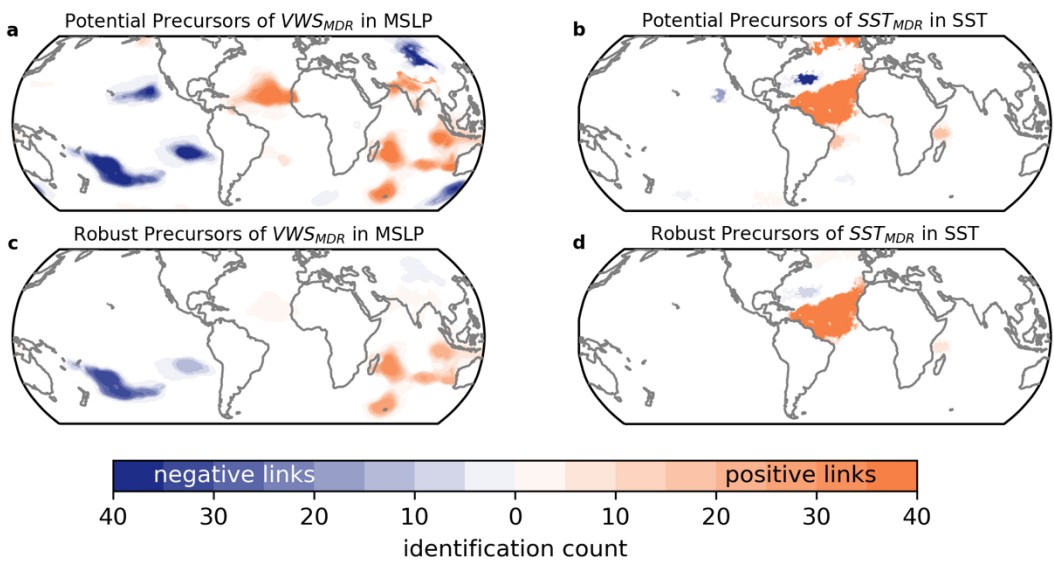

**Figure 7: Potential and robust precursors of VWS_{MDR} and SST_{MDR} in March. As Figure 4 but in March and with MSLP precursors for VWS_{MDR}.**

As expected, the overall skill of a hindcast based on these March precursors is lower than for the May hindcast but still considerable with a spearman rank correlation of 0.27 between the observed and the hindcasted ACE (Fig. 8). Despite being relatively low, this correlation is promising since this correlation is near zero in most operational forecast models (compare Klotzbach et al. 2017; Klotzbach 2019).

The linear model shows skill in hindcasting above median as well as extremely active seasons (Fig. 8b). For instance, an above 255 66th percentile season can be hindcasted with a true positive rate of 65% and a false positive rate of only 27%.

The hindcasts of the logistic regression model for above 66th percentile seasons has skill over a climatological forecast ($BSS =$ 0.11). The reliability diagram (Fig. 9c) shows a rather flat curve with few data points with high observed frequencies. This means that when the model predicts high probabilites for above 66th percentile seasons a relatively high number of these events are false positives. For above median TC activity the reliability curve is substantially closer to the diagonal and the skill over 260 a climatological forecast is higher ($BSS =$ 0.17 Fig. 9a).

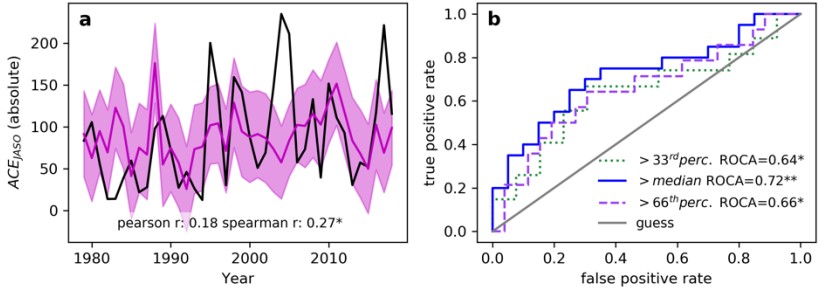

**Figure 8: Hindcast skill based on March reanalysis. As Figure 5 but for March.**

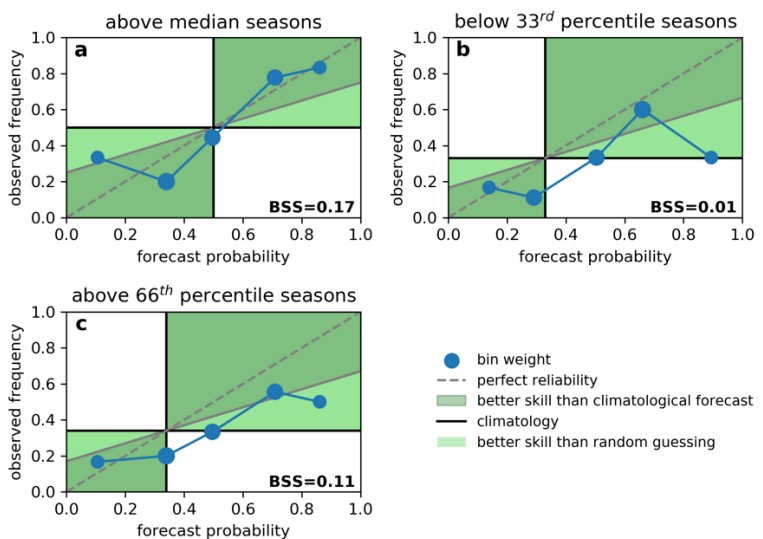

**Figure 9: Reliability diagrams for a logistic regression model on March precursors. As Figure 6 but for March.**

Our results suggest that the identified dipole pattern in MSLP in the southern Indian Ocean and the western south Pacific enhances VWS in the tropical Atlantic 4-6 months later (see Fig. 7c for the dipole) but the underlying mechanism is not obvious. We hypothesize that this dipole weakens the trade winds in the western Pacific, thereby favoring the formation of El Niño events. This would sub-sequentially lead to strong VWS in the Atlantic (during the main hurricane season).

We test this hypothesis by constructing a causal effect network. As input data we include time-series constructed as the MSLP difference between the southern Indian Ocean and the western south Pacific ("Delta MSLP"), the strength of the trade winds in the western Pacific ("Trade winds 850 hPa") and SSTs in the Niño 3.4 region ("SST Niño 3.4") (all regions are displayed in Figure 10a). The CEN is then calculated for the months of February to July which is roughly the period for which we want to test the hypothesis.

The detected causal links between the actors are shown in a Figure 10b. Indeed, weak trade winds in the western Pacific are suggested to favor the formation of El Niño events in the next month. Furthermore, a strong pressure gradient towards the Indian Ocean weakens the trade winds (on a time scale of two months). At the same time, strong trade winds increase the pressure difference between Indian Ocean and Pacific. Overall, although a more detailed analysis is needed, our analysis suggests that the identified March dipole, might indeed be physically linked to the upcoming Atlantic hurricane activity.

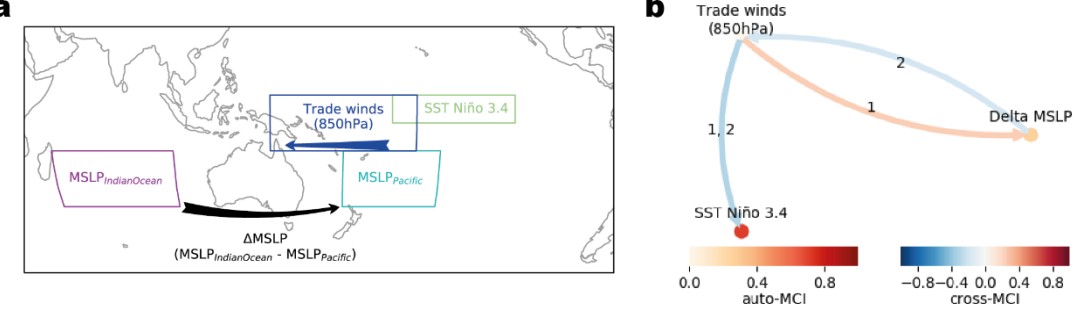

**Figure 10: Causal effect network (CEN) to test if the detected robust precursor in March affects ENSO variability. a) Regions included to construct the time-series that enter the CEN. To describe the robust precursors, we include the MSLP difference between the southern Indian Ocean (magenta box) and the western South Pacific (cyan box). We further include an index of western Pacific trade winds at 850hPa (blue box) and SSTs in the Nino3.4 region (green box). b) Resulting CEN of the time-series calculated over the regions as shown in a). For the CEN calculation only the months February to July and time lags of 1 to 4 months are considered. The color of the arrows indicates the link strengths, with the color of the nodes indicating the strength of the auto-dependence.**

The skill of the March forecast drops when detrended ACE anomalies are predicted using detrended precursor anomalies (see Fig. S3). One explanation for the reduced skill might be related to the simultaneous increase in ACE and the increase in Atlantic
SSTs over the period 1979-2018, partly modulated also by multi-decadal natural variability (Schleussner et al. 2013; Alexander et al. 2014). As the link between Atlantic SSTs and ACE is well-established (Murakami et al. 2018), and SSTs are auto-correlated on up to decadal timescales, such a detrending would be expected to reduce the skill of our model. Also, the linear trends of March precursors are small compared to the interannual variability and are not statistically significant (see Fig. S4). While a longer time series would be required to fully establish the independence of our findings from simultaneous trends in
the predictors and predictands, we are confident that our identified precursors of $VWS_{MDR}$ and $SST_{MDR}$ do indeed contain physically meaningful information.

Note that the identified robust precursors of strong $VWS_{MDR}$ over the southern hemispheric oceans might suffer from quality deficits of the reanalysis product, as only relatively few observations of SSTs and MSLP are available in these regions. We therefore perform a number of sensitivity tests to investigate how robust this signal is and whether it could be an artefact of
the used ERA5 reanalysis data. To do this, we conduct the same analysis using JRA55 reanalysis and obtain similar precursors in May and March (Fig. S5 & Fig. S7). Overall these precursors are yet less robust in JRA55 and the forecast skill is slightly reduced (Fig. S6 & Fig. S8).

The reason for this difference might also lie in the method applied to identify potential precursors (step 2 in Fig. 1). Applying the same clustering algorithm with the same parameters on a dataset with a different grid size leads to a minimally different
clustering behavior and might therefore affect the whole model building approach. The strong influence of the clustering step on the identified potential precursors and all the subsequent steps in the analysis could thus partially explains the differences between the results obtained with JRA55 and ERA5.

Finally, we test whether our model has skill outside of the period used for the main analysis (1979-2018) by applying a forecast model trained on 1980-2018 to the early period of JRA55 (1958-1978). In the pre-1979 period, our model captures main
features as a reduced hurricane activity in the 1970s after higher activity in the 1960s (see Fig. S9). It, however, systematically overestimates hurricane activity and the skill is lower than in the cross-validated hindcast of the period 1979-2018.

The reduced skill in the pre-1979 period could be a result of non-stationarities in precursors of Atlantic hurricane activity. For instance, changes in anthropogenic aerosol emissions lead to a suppression of tropical cyclone activity in the period 1950-1980 (Dunstone et al. 2013). This could explain the systematic overestimation of hurricane activity in our model as it does not
capture the effect of aerosols.

It has to be noted as well that reanalysis for time periods before the use of satellites (before 1979) are subject to considerable uncertainties especially in the southern hemisphere (Tennant 2004). It therefore remains difficult to investigate whether the identified relationship between March precursors and hurricane activity is robust under different climate states with the available datasets.

In summary, the identified MSLP precursors in March appear to be less robust than the well-documented May precursors. However, as far as it can be assessed with the given reanalysis datasets, sensitivity tests suggest that the identified March precursors indeed contain useful information contributing to a skillful seasonal forecast of ACE.

## 4 Discussion

A crucial component of statistical forecasting is the selection of meaningful predictors. Because too many included features

quickly lead to overfitting, methods are required to sub-select relevant predictors from a large set of potential predictors (Hawkins 2004). Here we showed that causal effect networks (CEN), a data-driven method based on causal inference techniques, can be used to identify robust predictors of a variable of interest.

Using CEN, we identified warm SSTs in the Atlantic and La Niña conditions in May as robust precursors of an active hurricane seasons in July-October. These precursors are consistent with the prevailing literature and thus show the usefulness of our

approach. We performed hindcasts based on these precursors and showed that the skill of our forecast model compares well with operational forecast models (Klotzbach et al. 2019), although, the real forecasting skill of our model can only be evaluated in the coming years.

At longer lead times, the skill of operational forecast models issued at the beginning of April is limited (Klotzbach et al. 2019). Here we also identified robust precursors in March including a region of Atlantic SSTs and two regions of mean sea level

pressure anomalies in the southern Indian Ocean and east of New Zealand. A model based on these precursors provides valuable hindcasts of above median and above $66^{th}$ percentile seasonal activity. We speculate on the involved mechanism at play, and suggest that a strong pressure gradient in that region weakens the trade winds in the western Pacific, which would favor the formation of El Niño events, which in turn are associated with reduced hurricane activity. We provided some evidence for this hypothesis by applying a simple CEN on the involved actors, but more research is needed to show the robustness of

this link.

In this study we searched for robust precursors of two well-known favorable conditions for TC formation and intensification, that is, warm SSTs and low VWS in the Atlantic main development region. Including more variables to the characterization of favorable conditions, such as relative humidity or upper troposphere temperatures could further increase the skill. It might, however, be challenging to incorporate these conditions in our current framework which was constructed using seasonally

aggregated data. For relative humidity in particular, variability on shorter time scales than SSTs or VWS might be relevant in this context.

Here we constructed causal effect networks for favorable conditions in the hurricane season and their potential precursors in SSTs or MSLP at a fixed time lag of two or four months. Yet, mechanisms on different time-scales and lags might also play a role and might further affect our results (Runge 2018). Overall, we cannot guarantee that the identified links are "truly causal".

The causal effect network approach rather helps to identify "the least spurious links" and therefore most robust precursors or

most skillful predictors. We stress that physical knowledge of the underlying mechanisms is essential to ensure a meaningful interpretation of our data-centric approach.

A challenge that we did not address here are potential non-stationarities regarding the detected robust precursors (as discussed in Fink et al. 2010; Caron et al. 2015). Such non-stationarities could lead to varying forecast skill. Given the limited time span for which reliable reanalysis datasets exist, this issue remains difficult. Applying our approach to climate simulations for which longer time series are available is therefore a logical next step.

Our here proposed technique to construct statistical forecasting models is generic and can easily be applied for other meteorological phenomena. It could for instance be applied to forecast seasonal hurricane activity in other basins for which fewer forecasts exist.

From a methodological viewpoint, the most sensitive step in the approach seems to be the identification of potential precursors (step 2 in Fig. 1). Depending on the choice of the free parameters of the clustering algorithm and significance threshold for correlated grid-cells, the detected potential precursor regions can vary and subsequently affect the causal network. Improving the robustness of this step or finding alternative ways of defining these potential precursors would further enhance the applicability of the method. Given the recent advances in novel machine learning techniques we are confident that this method can be further improved.

**5 Conclusions**

Using a causal effect network approach, we identified skillful spring predictors of seasonal Atlantic hurricane activity from July to October. For shorter lead times of two months, the identified precursor regions represent well-documented physical drivers. Statistical forecast models based on these drivers yield considerable prediction skill, demonstrating the potential of our method. For longer lead times of up to four months, our method suggests a pressure dipole between the southern Indian Ocean and the western South Pacific as a predictor of hurricane activity in the following season. A prediction model based on these March precursors still shows skill, but challenges in predicting in particular highly active hurricane seasons remain.

We see different entry points for our findings to be incorporated into applied seasonal hurricane forecasts. Besides a direct application of our early April forecast model, we encourage other statistical forecasting groups to investigate whether our newly identified predictors can help to improve their statistical forecast models. Furthermore, the causal links identified here could form the basis for hybrid forecasting techniques where a dynamical forecast ensemble is constrained by selecting only members that adequately reproduce the causal links as demonstrated by Dobrynin et al. (2018).

Improved seasonal forecasting with long lead-times can support seasonal planning of disaster risk reduction measures, particularly also related to disaster relief and emergency aid provision. While basin scale dissipated energy does not directly provide risk profiles for individual countries, it allows to inform decision making on the regional level, including on financial support needs pooled e.g. in the Caribbean Catastrophe Risk Insurance Facility serving Caribbean islands states (CCRIF). As such, improved seasonal forecasting can provide essential information to ensure hurricane preparedness in affected countries.

**Code availability**

All python scripts required to reproduce the analysis are available under:

github.com/peterpeterp/atlantic_ace_seasonal_forecast

**Author contributions**

All authors conceived the study. P.P. did the analysis and wrote the manuscript with contributions from all authors.

**Competing interests**

The authors declare no competing interests.


**Acknowledgements**

We acknowledge the openly available TIGRAMITE tool developed by Jakob Runge (github.com/jakobrunge/tigramite). We also acknowledge the IBTrACS dataset (ncdc.noaa.gov/ibtracs), fifth generation of ECMWF atmospheric reanalyses (ecmwf.int/en/forecasts/datasets/reanalysis-datasets/era5) and the Japanese 55-year reanalysis (jra.kishou.go.jp/JRA-55). We

thank the anonymous reviewers whose comments helped improve this manuscript. P.P. and C.-F.S. acknowledge support by the German Federal Ministry of Education and Research (01LN1711A). T.G. acknowledges from the German Federal Ministry of Education and Research (BMBF) under the research project SLICE (01LA1829A). M.K. was supported by the ERC Advanced Grant "ACRCC" (Grant 339390) and received funding from the European Union's Horizon 2020 research and innovation programme under the Marie Skłodowska-Curie grant agreement (No 841902).

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
