# Peer review of "Robust Predictors for Seasonal Atlantic Hurricane Activity Identified with Causal Effect Networks"

_Weather and Climate Dynamics, 2020_

## Referee Comment (RC1) · Anonymous Referee #1 · 8 Apr 2020

The study introduces a statistical forecast model of the July-October North Atlantic accumulated cyclone energy (ACE) wherein the causal effect network (CEN) approach is used to identify relevant precursors of the seasonal tropical cyclone (TC) activity in late and early spring of the same year. In the current version, I see the paper largely as a demonstration of the CEN method. There is not much value in other results. In the case of May forecasts, the robust precursors are well known. Results of the March forecasts are more interesting. However, it is not clear what physical mechanisms are at play, and whether the reported results will hold if a longer observational record is used. To improve the manuscript, I suggest the following: a) Provide more discussion of the physical relevance of the March precursors. Incidentally, does the MSLP dipole

pattern also exhibit a trend? b) Build a forecasting model based on March reanalysis data using a longer observational record. At least JRA55 data is available since 1958. This would allow to assess the robustness of the presented results and perhaps provide more insight into the physical mechanisms.

Minor Comments: 1. Line 30: I suggest replacing "regional" with "global and regional". 2. Line 65: What time period reanalyses data do you use? 3. Line 90: Please clarify why for the seasonal forecasting case potentially existing common drivers do not represent a problem. 4. Line 105: Maybe "for seasonal hurricane activity of the same year"? 5. Line 110: I suggest providing more background why SST and MSLP are used to find precursors of VWS. 6. Line 230: "(Fig. 7c)" instead of "(Fig. 7b)". 7. Line 235: "(see Figs. 7b and d)" instead of "(see Fig. 7c and d)". 8. Line 290: Please clarify why it may be challenging to incorporate other favorable conditions into your framework.

―――――――――――――――――

---

## Referee Comment (RC2) · Anonymous Referee #2 · 9 Apr 2020

**General comments**

The authors are using a causal effect network to forecast Atlantic tropical cyclone activity (July-October) as presented by accumulated cyclone energy (ACE). They developed two models, using ERA5 and JRA-55 data, for the months of March and May. Analysing the deterministic and probabilistic skill of their models, they find that it is competitive with other seasonal forecasts currently available.

The technique used here identifies predictors that have already been recognized in previous studies. So in that sense, the manuscript doesn't lead to new insights into the drivers of Atlantic cyclone activity. However, the objective of the paper is more to showcase this new method and the fact that it recovers the known drivers lends credibility to the results. I would encourage the authors to apply this technique for basins which have received less attention and for which the current forecast systems have more difficulty (e.g. Australian basin).

As I mention in my comments below, the authors identify regions of mlsp in the southern hemisphere In March as robust predictors of vertical wind shear during the hurricane season. Given the limited amount of observations that go into constructing the reanalyses over that region, I was wondering whether the authors thought this link was real or an artifact of the lack of observations. The authors should provide a comment to that effect in the manuscript.

Finally, the text is well written and easy to follow. I recommend it for publication after the minor points below have been addressed.

**Specific comments**

Line 18: "Earlier seasonal hurricane forecasting provides a multi-month lead time to implement more effective disaster risk reduction measures."

I'm not aware of any organization or government using seasonal forecasts for disaster risk reduction. Are the authors aware of any? If not, I would recommend removing this sentence from the abstract.

Line 26: "Preparedness for the secondary impacts can however be improved if reliable forecasts for the potential risks of the upcoming hurricane season are available (Martinez 2018)."

Does this refer to seasonal forecast? If so, it might be worth adding a sentence explaining how these forecasts are used in that context.

Line 156: "The color of the nodes indicates the strength of the auto dependence,"

auto dependence has not been defined. Can the authors explain what it means? And how is the strength of the link defined?

Line 182: "our cross-validated forecast seems competitive with operational forecasts".

Could the authors be more precise? Which operational forecasts are they referring to?

Line 199: "deficit to predict some of the most active seasons might be due to missing relevant predictors"

There is also a stochastic component to TC formation. Two different years with similar large-scale fields conditions would/could lead to a different numbers of cyclones.

Line 231: "As robust precursors, a high-pressure system over the southern Indian Ocean and a low-pressure system eastward of New Zealand are identified in nearly all training sets"

Is this a true feature or possibly a feature of the reanalysis, which have very little observation over the southern ocean?

Line 262: "Overall these precursors seem less robust in JRA55 and thereby the forecast skill is also slightly reduced"

The Spearman correlation is higher using JRA-55 in March actually.

**Technical corrections**

Line 21: "Tropical cyclones (TCs) are among the most damaging weather events in many tropical and subtropical regions."

This statement should be referenced.

Line 28: Klotzbach (2019) should be Klotzbach et al. 2019

Klotzbach, P. J., E. S. Blake, J. Camp, L.-P. Caron, J. Chan, N. Kang, Y. Kuleshov, S.-M. Lee, H. Murakami, M. Saunders, Y. Takaya, F. Vitart, and R. Zhan, 2019: Seasonal tropical cyclone forecasting. Tropical Cyclone Research and Review, 8, 134-149, doi: 10.6057/2019TCRR03.03.

Line 30: "A whole variety of forecasting methods are applied ranging from purely statistical forecasts to forecasts based on regional climate model simulations and hybrid approaches."

I'm not familiar with the methodology of every group, but nowadays global climate models are used instead of regional climate models.

Line 32: "Their skill depends on their ability to represent TC genesis and development and their capacity to forecast the large-scale circulation over the Atlantic main development region (MDR)."

As well as their ability to adequately represent the interaction between the two.

Line 35: "With increasing spatial resolution their representation of TCs improves."

I would add a reference here.

Line 61: "official WMO agencies"

Line 64: "We use the monthly reanalysis data provided on a regular 1-degree grid."

Aren't the ERA5 data at 35 km resolution?

Line 88: "As such, we cannot exclude potential common drivers on longer, e.g. annual time scales"

Do you mean multi-annual or decadal time scales?

Line 99: "A statistical model is built"

Line 126: "we will still refer to our cross-validated predictions as "forecasts""

I would recommend using hindcast, to avoid confusion.

Line 199:  hypothise  -> hypothesize

Line 202: "as a predictor"

Line 219: "(BSS) is indicated in the lower right corner of each panel."

Line 294: "causal effect network rather helps to identify "the least spuriousl link"

Line 297: "The detected causal links might not be stationary over time"

Nonstationarity in the climate influence on TC activity has been pointed out by:

Fink AH, Schrage JM, Kotthaus S (2010) On the potential causes of the nonstationary correlations between West African precipitation and Atlantic Hurricane activity. J Clim 23(20):5437–5456

Caron, L-P, M Boudreault and C Bruyère (2015)  Changes in large-scale controls of Atlantic tropical cyclone activity with the phases of the Atlantic Multidecadal Oscillation. Climate Dynamics, 44, 1801-1821. doi:10.1007/s00382-014-2186-5.

Figure 1: I would like to thank the authors for taking the time to produce this figure. It helped a lot in understanding the methodology.

---

## Author Comment (AC1) · 28 May 2020

The study introduces a statistical forecast model of the July-October North Atlantic accumulated cyclone energy (ACE) wherein the causal effect network (CEN) approach is used to identify relevant precursors of the seasonal tropical cyclone (TC) activity in late and early spring of the same year. In the current version, I see the paper largely as a demonstration of the CEN method. There is not much value in other results. In the case of May forecasts, the robust precursors are well known. Results of the March forecasts are more interesting. However, it is not clear what physical mechanisms are at play, and whether the reported results will hold if a longer observational record is used.

> *We thank the reviewer for her/his useful comments. We agree that in the first version of the manuscript the discussion of the March precursors lacked some depth. As discussed below in more detail, we therefore added some analysis according to the reviewer`s suggestions.*

To improve the manuscript, I suggest the following:

a) Provide more discussion of the physical relevance of the March precursors.

> *In the new version of the manuscript, (see Fig. 10 and lines 268-282) we now hypothesize on the mechanism connecting the detected dipole precursor pattern in March to enhanced VWS in July-August. More precisely, we speculate that a high-pressure anomaly in the southern Indian Ocean and a low-pressure anomaly in the western South Pacific could weaken the trade winds in the western Pacific which is favorable for El Niño formation (which in turn is known to have an enhancing influence on VWS in the Atlantic).*
> *We test this hypothesis by constructing a causal effect network that shows the links between SSTs in the El Niño 3.4 region, trade winds in the western Pacific and the strength of the identified MSLP dipole. This analysis supports our hypothesis. Note, however, that more research is needed to confirm the robustness of this linkage, something we also clearly state in the paper.*

Incidentally, does the MSLP dipole pattern also exhibit a trend?

> *We use detrended time series to identify precursors and therefore do not expect the link between precursors and predictands to rely on such trends. The reviewer is right, however, that the skill in our final forecast model could depend on trends in the precursors and we thank him/her for pointing this out. We have, therefore, additionally analyzed the linear trends in three regions that have been identified as robust precursors in most training sets (see Figure S3).*

> *For the two identified MSLP precursors, no significant trend is detected (p-values 0.16 for the western Pacific precursor and 0.67 for the Indian Ocean precursor). Also, the trend in the Atlantic SST precursor is insignificant (p-value=0.25). While we cannot rule out that (insignificant) trends maybe contribute to the prediction skill, we conclude that they cannot fully explain the relationships identified and are thus not primary responsible for our model skill.*

> *We added a discussion about the potential influence of trends in line 293-296.*

b) Build a forecasting model based on March reanalysis data using a longer observational record. At least JRA55 data is available since 1958. This would allow to assess the robustness of the presented results and perhaps provide more insight into the physical mechanisms.

> *We thank the reviewer for this suggestion. In the modified version of the manuscript, we now test the skill of a prediction model (that is trained over the period 1980-2018) to the earlier period of JRA55 (1958-1978). The skill of our model is reduced in the pre-1979 period which is mainly due to a systematic overestimation in hurricane activity.*
> *We hypothesize that this overestimation could be explained by the influence of anthropogenic aerosols on hurricane activity that is not captured in our model. In the period 1950-1980 high aerosol emissions lead to tropical cyclone suppression in the Atlantic. Regulations in the late 1970s lead to fewer emissions weakening the tropical cyclone suppression* (Dunstone et al. 2013)*. We also think that the quality of reanalysis products before the use of satellites before 1979 introduce systematic uncertainties which make it extremely difficult to further investigate the robustness of the our results (compare* Tennant 2004)*. We nevertheless included this sensitivity test into the SI (Fig. S8) and discuss it in line 308-319 of the manuscript.*

Minor Comments:

1. Line 30: I suggest replacing "regional" with "global and regional".

> *We thank the reviewer for spotting this error in the manuscript and we now changed "regional" to "global".*

2. Line 65: What time period reanalyses data do you use?

> *We added the periods used for both datasets in the manuscript (line 67 and 69-70).*

3. Line 90: Please clarify why for the seasonal forecasting case potentially existing common drivers do not represent a problem.

> *We agree with the reviewer that this part was not clear enough. We reformulated this paragraph and only explain the shortcomings of our application (line 89-95). We also discuss these shortcomings again in the discussion section (line 347-352).*

4. Line 105: Maybe "for seasonal hurricane activity of the same year"?

> *We modified the manuscript accordingly.*

5. Line 110: I suggest providing more background why SST and MSLP are used to find precursors of VWS.

> *We agree with the reviewer, that in the initial version of the manuscript an explanation was lacking. We added a sentence motivating the choice of these variables in line 115-117.*

6. Line 230: "(Fig. 7c)" instead of "(Fig. 7b)".

*Done*

7. Line 235: "(see Figs. 7b and d)" instead of "(see Fig. 7c and d)".

*Done*

8. Line 290: Please clarify why it may be challenging to incorporate other favorable conditions into your framework.

*We agree with the reviewer that this statement should be better explained. Although relative humidity is an important variable for the analysis of tropical cyclone formation it varies on shorter time scales and seasonal averages of relative humidity might not be insightful. SSTs and VWS are both to some extent modulated by ENSO and seasonal averages can therefore differ substantially. We added an explaining sentence in line 343-346.*

**References**

Dunstone NJ, Smith DM, Booth BBB, et al (2013) Anthropogenic aerosol forcing of Atlantic tropical storms. Nat Geosci 6:534–539. doi: 10.1038/ngeo1854

Tennant W (2004) Considerations when using pre-1979 NCEP/NCAR reanalyses in the southern hemisphere. Geophys Res Lett 31:n/a-n/a. doi: 10.1029/2004GL019751

---

## Author Comment (AC2) · 28 May 2020

General comments

The authors are using a causal effect network to forecast Atlantic tropical cyclone activity (July-October) as presented by accumulated cyclone energy (ACE). They developed two models, using ERA5 and JRA-55 data, for the months of March and May. Analysing the deterministic and probabilistic skill of their models, they find that it is competitive with other seasonal forecasts currently available.

The technique used here identifies predictors that have already been recognized in previous studies. So in that sense, the manuscript doesn't lead to new insights into the drivers of Atlantic cyclone activity. However, the objective of the paper is more to showcase this new method and the fact that it recovers the known drivers lends credibility to the results. I would encourage the authors to apply this technique for basins which have received less attention and for which the current forecast systems have more difficulty (e.g. Australian basin).

> *We thank the reviewer for this useful suggestion. We think that for the present manuscript it makes sense to keep the focus on the Atlantic basin, exactly because the involved mechanisms are relatively well documented. Thus, our study should serve as a proof of concept of the method and we intend to apply it to other basins in a follow up study (see line 358-359). We have modified the framing of the manuscript accordingly.*

As I mention in my comments below, the authors identify regions of mlsp in the southern hemisphere In March as robust predictors of vertical wind shear during the hurricane season. Given the limited amount of observations that go into constructing the reanalyses over that region, I was wondering whether the authors thought this link was real or an artifact of the lack of observations. The authors should provide a comment to that effect in the manuscript.

> *We share the concern raised by the reviewer and we are aware of the possibility of detecting artifacts of the reanalysis as precursors for our model. In the revised manuscript we therefore discuss a potential mechanism that could explain the influence of the identified MSLP dipole in March on VWS in the hurricane season (see Fig. 10 and lines 268-281).*

> *We speculate that a high-pressure anomaly in the southern Indian Ocean and a low-pressure anomaly in the western South Pacific could weaken the trade winds in the western Pacific which is favorable for El Nino formation (which in turn is known to have an enhancing influence on VWS in the Atlantic).*
> *We test this hypothesis by constructing a causal effect network that shows the links between SSTs in the El Nino3.4 region, trade winds in the western Pacific and the strength of the identified MSLP dipole. This analysis supports our hypothesis. Note, however, that more research is needed to confirm the robustness of this linkage, something we also clearly state in the paper.*

> *Furthermore, we extended the sensitivity tests adding a trend analysis for the identified precursors (see Fig. S3, line 293-296) and an application of a forecast model trained on the period of 1980-2018 to an earlier test period (1958-1978) using the JRA55 reanalysis (see Fig. S8, line 308-319 and the response to reviewer 1).*

*Given the limited available datasets, we cannot fully rule out the eventuality of detecting artifacts as precursors. However, our sensitivity tests suggest that the identified MSLP precursors have a physical link to VWS in during the hurricane season.*

Finally, the text is well written and easy to follow. I recommend it for publication after the minor points below have been addressed.

*We thank the reviewer for this overall positive feedback.*

**Specific comments**

Line 18: "Earlier seasonal hurricane forecasting provides a multi-month lead time to implement more effective disaster risk reduction measures."

I'm not aware of any organization or government using seasonal forecasts for disaster risk reduction. Are the authors aware of any? If not, I would recommend removing this sentence from the abstract.

*With this statement we wanted to refer to long term planning of governments and financial institutions. As we don't have a specific reference outlining how seasonal forecasts are used in this context we removed the sentence.*

Line 26: "Preparedness for the secondary impacts can however be improved if reliable forecasts for the potential risks of the upcoming hurricane season are available (Martinez 2018)."

Does this refer to seasonal forecast? If so, it might be worth adding a sentence explaining how these forecasts are used in that context.

*We thank the reviewer for this comment. Martinez 2018 indeed refers to forecasts on shorter time scales. We now reference (Murphy et al. 2001) that discusses the use of seasonal forecasts more generally. We think that for secondary impacts the statements of this study are also applicable to seasonal hurricane forecasts.*

Line 156: "The color of the nodes indicates the strength of the auto dependence,"

auto dependence has not been defined. Can the authors explain what it means? And how is the strength of the link defined?

*We thank the reviewer for this comment. We used the term "auto-dependence" to differentiate it from "auto-correlation". Thus, it refers to how much the past of a process influences the next time-step. Both auto-dependence and link strength are calculated using partial correlations as described in Runge et al. (2019).*
*We only use the link strength to identify robust precursors. The final forecast model is based on a simple regression which is why we do not think that in depth discussion of the interpretation of link strengths is necessary.*
*In the revised manuscript we shortly explain the meaning of auto-dependence and link strength in the caption of Fig. 3 (line 162-165).*

Line 182: "our cross-validated forecast seems competitive with operational forecasts". Could the authors be more precise? Which operational forecasts are they referring to?

*We were referring to Fig. 1 from Klotzbach et al. 2019 in which the skill of seasonal April forecasts of ACE provided by CSU, TSR and NOAA is presented. We added the abbreviations of these institutes to the manuscript and specifically mention Fig. 1 in the citation (line 191-192).*

Line 199: "deficit to predict some of the most active seasons might be due to missing relevant predictors"

There is also a stochastic component to TC formation. Two different years with similar large-scale fields conditions would/could lead to a different numbers of cyclones.

*We thank the reviewer for this comment. We added the following sentence to the manuscript (line 212-213):*
*"Furthermore, it has to be noted that there is some stochastic component to TC formation which systematically limits the skill of our empirical forecast model that is based on favorable conditions for TC formation."*

Line 231: "As robust precursors, a high-pressure system over the southern Indian Ocean and a low-pressure system eastward of New Zealand are identified in nearly all training sets"

Is this a true feature or possibly a feature of the reanalysis, which have very little observation over the southern ocean?

*We fully understand the concern of the reviewer. Please find our answer to this question below general remarks of the reviewer.*

Line 262: "Overall these precursors seem less robust in JRA55 and thereby the forecast skill is also slightly reduced"

The Spearman correlation is higher using JRA-55 in March actually.

*We thank the reviewer for pointing this out. We think, however, that there has been some confusion. The Spearman correlation in March is 0.22 for JRA55 and 0.27 for ERA5.*

Technical corrections

Line 21: "Tropical cyclones (TCs) are among the most damaging weather events in many tropical and subtropical regions."

This statement should be referenced.

*We included a reference to the MunichRe natural hazard database* (Munich Re 2020)*: https://natcatservice.munichre.com/*

Line 28: Klotzbach (2019) should be Klotzbach et al. 2019

Klotzbach, P. J., E. S. Blake, J. Camp, L.-P. Caron, J. Chan, N. Kang, Y. Kuleshov, S.-M. Lee, H. Murakami, M. Saunders, Y. Takaya, F. Vitart, and R. Zhan, 2019: Seasonal tropical cyclone forecasting. Tropical Cyclone Research and Review, 8, 134-149, doi: 10.6057/2019TCRR03.03.

*Done*

Line 30: "A whole variety of forecasting methods are applied ranging from purely statistical forecasts to forecasts based on regional climate model simulations and hybrid approaches."

I'm not familiar with the methodology of every group, but nowadays global climate models are used instead of regional climate models.

*We thank the reviewer for correcting this statement. We changed "regional" to "global".*

Line 32: "Their skill depends on their ability to represent TC genesis and development and their capacity to forecast the large-scale circulation over the Atlantic main development region (MDR)."

As well as their ability to adequately represent the interaction between the two.

*We fully agree with the reviewer and added the sentence (line 37).*

Line 35: "With increasing spatial resolution their representation of TCs improves." I would add a reference here.

*We added the following reference* (Roberts et al. 2020)*:*
*1. Roberts, M. J. et al. Impact of model resolution on tropical cyclone simulation using the HighResMIP-PRIMAVERA multimodel ensemble. J. Clim. 33, 2557–2583 (2020).*

Line 61: "official WMO agencies"

*Done*

Line 64: "We use the monthly reanalysis data provided on a regular 1-degree grid."

Aren't the ERA5 data at 35 km resolution?

*ERA5 is indeed available on a higher resolution. We are using the 1-degree grid (which is also provided on the website). We hope that removing the word "provided" clarifies this.*

Line 88: "As such, we cannot exclude potential common drivers on longer, e.g. annual time scales"

Do you mean multi-annual or decadal time scales?

*We meant multi-annual up to multi-decadal time scales. Based on a comment of reviewer 1, we rewrote this paragraph also specifying these time scales (line 89-95).*

Line 99: "A statistical model is built"

*Thanks*

Line 126: "we will still refer to our cross-validated predictions as "forecasts""

I would recommend using hindcast, to avoid confusion.

*We agree with the reviewer and write about "hindcasts" throughout the revised manuscript.*

Line 199: hypothise -> hypothesize

*Done*

Line 202: "as a predictor"

*Done*

Line 219: "(BSS) is indicated in the lower right corner of each panel."

*Done*

Line 294: "causal effect network rather helps to identify "the least spuriously link"

*Done*

Line 297: "The detected causal links might not be stationary over time"

Nonstationarity in the climate influence on TC activity has been pointed out by:

Fink AH, Schrage JM, Kotthaus S (2010) On the potential causes of the nonstationary correlations between West African precipitation and Atlantic Hurricane activity. J Clim 23(20):5437–5456

Caron, L-P, M Boudreault and C Bruyère (2015) Changes in large-scale controls of Atlantic tropical cyclone activity with the phases of the Atlantic Multidecadal Oscillation. Climate Dynamics, 44, 1801-1821. doi:10.1007/s00382-014-2186-5.

*We thank the reviewer for suggesting these references (that we now included in line 93, 354). Following the suggestion of reviewer 1, we also included a test using the earlier period (1958-1978) of JRA55 (see Fig. S8). Since the data quality of the reanalysis before the satellite era (before 1979) seems questionable in the southern hemispheric oceans (Tennant 2004) we still cannot fully investigate potential non-stationarities in our precursors. See line 308-319. Nevertheless, we have added a sentence discussing this possibility.*

Figure 1: I would like to thank the authors for taking the time to produce this figure. It helped a lot in understanding the methodology.

*We would like to thank the author for this nice comment. We are very glad to hear that this figure is helpful.*

**References**

Munich Re (2020) Natural catastrophe know-how for risk management and research. https://natcatservice.munichre.com. Accessed 19 May 2020

Murphy SJ, Washington R, Downing TE, et al (2001) Seasonal forecasting for climate hazards: Prospects and responses. Nat Hazards 23:171–196. doi: 10.1023/A:1011160904414

Roberts MJ, Camp J, Seddon J, et al (2020) Impact of model resolution on tropical cyclone simulation using the HighResMIP-PRIMAVERA multimodel ensemble. J Clim 33:2557–2583. doi: 10.1175/JCLI-D-19-0639.1

Runge J, Nowack P, Kretschmer M, et al (2019) Detecting and quantifying causal associations in large nonlinear time series datasets. Sci Adv 5:eaau4996. doi: 10.1126/sciadv.aau4996

Tennant W (2004) Considerations when using pre-1979 NCEP/NCAR reanalyses in the southern hemisphere. Geophys Res Lett 31:n/a-n/a. doi: 10.1029/2004GL019751

---

## Author Response (AR2)

**Response to Anonymous Reviewer #1**

My original comments have been adequately addressed, and the manuscript is essentially ready for publication. I only have one suggestion, as follows:

1. I would add Lines 251-253 from the original version of the manuscript (regarding the forecast skill when detrended ACE anomalies are predicted) after Line 296 in the revised version (version 3). I believe this information is omitted from the revision. In my opinion, it is quite important and adds to the interpretation of the March forecasts.

> We agree with the reviewer, that the forecast skill for detrended ACE based on detrended predictors contains important information. In the revised version we added Figure S3 back to the SI and reformulated lines 298-302 to include the information of lines 251-253 from the initial submission, as suggested by the reviewer.

**Response to Anonymous Reviewer #2**

This is my second time reviewing this manuscript. All my initial queries have been answered and I recommend this manuscript for publication.

I have also identified three very minor mistakes and have one suggestion for the authors (all listed below).

Line 64:I think "agencies" should be singular in this case, since there is only one agency covering the Atlantic basin.
> We changed "WMO agencies" to "WMO agency" as suggested.

Line 278: Replace "Further" with "Furthermore"
> Done

p15. Conclusion should be Section 5, not 4.
> Thanks!

For Figure 5b and 8b, I'm having trouble differentiating between the 33th perc. and 66th perc. Could be my printer, but I would still recommend picking colors that are more distinct.
> We agree, that the color choice wasn't ideal. We changed the colors and replaced one dashed line by a stippled line such that every line has a different style and color.